# Gradient-based Adaptive Markov Chain Monte Carlo

**Michalis K. Titsias**
DeepMind
London, UK
mtitsias@google.com

**Petros Dellaportas**
Department of Statistical Science
University College of London, UK
Department of Statistics, Athens
Univ. of Econ. and Business, Greece
and The Alan Turing Institute, UK

## Abstract

We introduce a gradient-based learning method to automatically adapt Markov chain Monte Carlo (MCMC) proposal distributions to intractable targets. We define a maximum entropy regularised objective function, referred to as generalised speed measure, which can be robustly optimised over the parameters of the proposal distribution by applying stochastic gradient optimisation. An advantage of our method compared to traditional adaptive MCMC methods is that the adaptation occurs even when candidate state values are rejected. This is a highly desirable property of any adaptation strategy because the adaptation starts in early iterations even if the initial proposal distribution is far from optimum. We apply the framework for learning multivariate random walk Metropolis and Metropolis-adjusted Langevin proposals with full covariance matrices, and provide empirical evidence that our method can outperform other MCMC algorithms, including Hamiltonian Monte Carlo schemes.

## 1 Introduction

Markov chain Monte Carlo (MCMC) is a family of algorithms that provide a mechanism for generating dependent draws from arbitrarily complex distributions. The basic set up of an MCMC algorithm in any probabilistic (e.g. Bayesian) inference problem, with an intractable target density $\pi(x)$, is as follows. A discrete time Markov chain $\{X_t\}_{t=0}^{\infty}$ with transition kernel $P_{\theta}$, appropriately chosen from a collection of $\pi$-invariant kernels $\{P_{\theta}(\cdot, \cdot)\}_{\theta \in \Theta}$, is generated and the ergodic averages $\mu_t(F) = t^{-1} \sum_{i=0}^{t-1} F(X_i)$ are used as approximations to $E_{\pi}(F)$ for any real-valued function $F$. Although in principle this sampling setup is simple, the actual implementation of any MCMC algorithm requires careful choice of $P_{\theta}$ because the properties of $\mu_t$ depend on $\theta$. In common kernels that lead to random walk Metropolis (RWM), Metropolis-adjusted Langevin (MALA) or Hamiltonian Monte Carlo (HMC) algorithms the kernels $P_{\theta}$ are specified through an accept-reject mechanism in which the chain moves from time $t$ to time $t + 1$ by first proposing candidate values $y$ from a density $q_{\theta}(y|x)$ and accepting them with some probability $\alpha(x_t, y)$ and setting $x_{t+1} = y$, or rejecting them and setting $x_{t+1} = x_t$. Since $\theta$ directly affects this acceptance probability, it is clear that one should choose $\theta$ so that the chain does not move too slowly or rejects too many proposed values $y$ because in both these cases convergence to the stationary distribution will be slow. This has been recognised as early as in [22] and has initiated exciting research that has produced optimum average acceptance probabilities for a series of algorithms; see [30, 31, 32, 15, 6, 8, 34, 7, 35, 9]. Such optimal average acceptance probabilities provide basic guidelines for adapting single step size parameters to achieve certain average acceptance rates.

More sophisticated adaptive MCMC algorithms that can learn a full set of parameters $\theta$, such as a covariance matrix, borrow information from the history of the chain to optimise some criterion reflecting the performance of the Markov chain [14, 5, 33, 13, 2, 1, 4]. Such methods are typically

non gradient-based and the basic strategy they use is to sequentially fit the proposal $q_\theta(y|x)$ to the history of states $x_{t-1}, x_t, \ldots$, by ignoring also the rejected state values. This can result in very slow adaptation because the initial Markov chain simulations are based on poor initial $\theta$ and the generated states, from which $\theta$ is learnt, are highly correlated and far from the target. The authors in [34] call such adaptive strategies 'greedy' in the sense that they try to adapt too closely to initial information from the output and take considerable time to recover from misleading initial information.

In this paper, we develop faster and more robust gradient-based adaptive MCMC algorithms that make use of the gradient of the target, $\nabla \log \pi(x)$, and they learn from both actual states of the chain and proposed (and possibly rejected) states. The key idea is to define and maximise w.r.t. $\theta$ an entropy regularised objective function that promotes high acceptance rates and high values for the entropy of the proposal distribution. This objective function, referred to as generalised speed measure, is inspired by the speed measure of the infinite-dimensional limiting diffusion process that captures the notion of speed in which a Markov chain converges to its stationary distribution [32]. We maximise this objective function by applying stochastic gradient variational inference techniques such as those based on the reparametrisation trick [19, 29, 40]. An advantage of our algorithm compared to traditional adaptive MCMC methods is that the adaptation occurs even when candidate state values are rejected. In fact, the adaptation can be faster when candidate values $y$ are rejected since then we make always full use of the gradient $\nabla \log \pi(y)$ evaluated at the rejected $y$. This allows the adaptation to start in early iterations even if the initial proposal distribution is far from optimum and the chain is not moving. We apply the method for learning multivariate RWM and MALA proposals where we adapt full covariance matrices parametrised efficiently using Cholesky factors. In the experiments we demonstrate our algorithms to multivariate Gaussian targets and Bayesian logistic regression and empirically show that they outperform several other baselines, including advanced HMC schemes.

## 2   Gradient-based adaptive MCMC

Assume a target distribution $\pi(x)$, known up to some unknown normalising constant, where $x \in \mathbb{R}^n$ is the state vector. To sample from $\pi(x)$ we consider the Metropolis-Hastings (M-H) algorithm that generates new states by sampling from a proposal distribution $q_\theta(y|x)$, that depends on parameters $\theta$, and accepts or rejects each proposed state by using the standard M-H acceptance probability

$$\alpha(x, y; \theta) = \min \left\{ 1, \frac{\pi(y) q_\theta(x|y)}{\pi(x) q_\theta(y|x)} \right\}. \tag{1}$$

While the M-H algorithm defines a Markov chain that converges to the target distribution, the efficiency of the algorithm heavily depends on the choice of the proposal distribution $q_\theta(x|y)$ and the setting of its parameters $\theta$.

Here, we develop a framework for stochastic gradient-based adaptation or learning of $q_\theta(x|y)$ that maximises an objective function inspired by the concept of speed measure that underlies the theoretical foundations of MCMC optimal tuning [30, 31]. Given that the chain is at state $x$ we would like: (i) to propose big jumps in the state space and (ii) accept these jumps with high probability. By assuming for now that the proposal has the standard random walk isotropic form, such that $q_\sigma(y|x) = \mathcal{N}(y|x, \sigma^2 I)$, the speed measure is defined as

$$s_\sigma(x) = \sigma^2 \times \alpha(x; \sigma), \tag{2}$$

where $\sigma^2$ denotes the variance, also called step size, of the proposal distribution, while $\alpha(x; \sigma)$ is the average acceptance probability when starting at $x$, i.e. $\alpha(x; \sigma) = \int \alpha(x, y; \sigma) q_\sigma(y|x) dy$. To learn a good value for the step size we could maximise the speed measure in Eq. 2, which intuitively promotes high variance for the proposal distribution together with high acceptance rates. In the theory of optimal MCMC tuning, $s_\sigma(x)$ is averaged under the stationary distribution $\pi(x)$ to obtain a global speed measure value $s_\sigma = \int \pi(x) s_\sigma(x) dx$. For simple targets and with increasing dimension this measure is maximised so that $\sigma^2$ is set to a value that leads to the acceptance probability $0.234$ [30, 31]. This subsequently leads to the popular heuristic for tuning random walk proposals: tune $\sigma^2$ so that on average the proposed states are accepted with probability $1/4$. Similar heuristics have been obtained for tuning the step sizes of more advanced schemes such as MALA and HMC, where $0.574$ is considered optimal for MALA [32] and $0.651$ for HMC [24, 9].

While the current notion of speed measure from Eq. 2 is intuitive, it is only suitable for tuning proposals having a single step size. Thus, in order to learn arbitrary proposal distributions $q_\theta(y|x)$,

where $\theta$ is a vector of parameters, we need to define suitable generalisations of the speed measure. Suppose, for instance, that we wish to tune a Gaussian with a full covariance matrix, i.e. $q_\Sigma(y|x) = \mathcal{N}(y|x, \Sigma)$. To achieve this we need to generalise the objective in Eq. 2 by replacing $\sigma^2$ with some functional $\mathcal{F}(\Sigma)$ that depends on the full covariance. An obvious choice is to consider the average distance $||y-x||^2$ given by the trace $\text{tr}(\Sigma) = \sum_i \sigma_i^2$. However, this is problematic since it can lead to learning proposals with very poor mixing. To see this note that since the trace is the sum of variances it can obtain high values even when some of the components of $x$ have very low variance, e.g. for some $x_i$ it holds $\sigma_i^2 \approx 0$, which can result in very low sampling efficiency or even non-ergodicity. In order to define better functionals $\mathcal{F}(\Sigma)$ we wish to exploit the intuition that for MCMC all components of $x$ need to jointly perform (relative to their scale) big jumps, a requirement that is better captured by the determinant $|\Sigma|$ or more generally by the entropy of the proposal distribution.

Therefore, we introduce a generalisation of the speed measure having the form,

$$s_\theta(x) = \exp\{\beta \mathcal{H}_{q_\theta(y|x)}\} \times \alpha(x; \theta) = \exp\{\beta \mathcal{H}_{q_\theta(y|x)}\} \times \int \alpha(x, y; \theta) q_\theta(y|x) dy, \qquad (3)$$

where $\mathcal{H}_{q_\theta(y|x)} = -\int q_\theta(y|x) \log q_\theta(y|x) dy$ denotes the entropy of the proposal distribution and $\beta > 0$ is an hyperparameter. Note that when the proposal distribution is a full Gaussian $q_\Sigma(y|x) = \mathcal{N}(y|x, \Sigma)$ then $\exp\{\beta \mathcal{H}_{q(y|x)}\} = \text{const} \times |\Sigma|^{\frac{\beta}{2}}$ which depends on the determinant of $\Sigma$. $s_\theta(x)$, referred to as *generalised speed measure*, trades off between high entropy of the proposal distribution and high acceptance probability. The hyperparameter $\beta$ plays the crucial role of balancing the relative strengths of these terms. As discussed in the next section we can efficiently optimise $\beta$ in order to achieve a desirable average acceptance rate.

In the following we make use of the above generalised speed measure to derive a variational learning algorithm for adapting the parameters $\theta$ using stochastic gradient-based optimisation.

## 2.1 Maximising the generalised speed measure using variational inference

During MCMC iterations we collect the pairs of vectors $(x_t, y_t)_{t>0}$ where $x_t$ is the state of the chain at time $t$ and $y_t$ the corresponding proposed next state, which if accepted then $x_{t+1} = y_t$. When the chain has converged each $x_t$ follows the stationary distribution $\pi(x)$, otherwise it follows some distribution that progressively converges to $\pi(x)$. In either case we view the sequence of pairs $(x_t, y_t)$ as non-iid data based on which we wish to perform gradient-based updates of the parameters $\theta$. In practice such updates can be performed with diminishing learning rates, or more safely completely stop after some number of burn-in iterations to ensure convergence. Specifically, given the current state $x_t$ we wish to take a step towards maximising $s_\theta(x_t)$ in Eq. 3 or equivalently its logarithm,

$$\log s_\theta(x_t) = \log \int \alpha(x, y; \theta) q_\theta(y|x_t) dy + \beta \mathcal{H}_{q_\theta(y|x_t)}. \qquad (4)$$

The second term is just the entropy of the proposal distribution, which typically will be analytically tractable, while the first term involves an intractable integral. To approximate the first term we work similarly to variational inference [18, 10] and we lower bound it using Jensen's inequality,

$$\log s_\theta(x_t) \geq \mathcal{F}_\theta(x_t) = \int q_\theta(y|x_t) \log \min \left\{ 1, \frac{\pi(y) q_\theta(x_t|y)}{\pi(x_t) q_\theta(y|x_t)} \right\} dy + \beta \mathcal{H}_{q_\theta(y|x_t)} \qquad (5)$$

$$= \int q_\theta(y|x_t) \min \left\{ 0, \log \frac{\pi(y)}{\pi(x_t)} + \log \frac{q_\theta(x_t|y)}{q_\theta(y|x_t)} \right\} dy + \beta \mathcal{H}_{q_\theta(y|x_t)}. \qquad (6)$$

To take a step towards maximising $\mathcal{F}_\theta$ we can apply standard stochastic variational inference techniques such as the score function method or the reparametrisation trick [11, 26, 28, 19, 29, 40, 20]. Here, we focus on the case $q_\theta(y|x_t)$ is a reparametrisable distribution such that $y = \mathcal{T}_\theta(x_t, \epsilon)$ where $\mathcal{T}_\theta$ is a deterministic transformation and $\epsilon \sim p(\epsilon)$. $\mathcal{F}_\theta(x_t)$ can be re-written as

$$\mathcal{F}_\theta(x_t) = \int p(\epsilon) \min \left\{ 0, \log \frac{\pi(\mathcal{T}_\theta(x_t, \epsilon))}{\pi(x_t)} + \log \frac{q_\theta(x_t|\mathcal{T}_\theta(x_t, \epsilon))}{q_\theta(\mathcal{T}_\theta(x_t, \epsilon)|x_t)} \right\} d\epsilon + \beta \mathcal{H}_{q_\theta(y|x_t)}.$$

Since MCMC at the $t$-th iteration proposes a specific state $y_t$ constructed as $\epsilon_t \sim p(\epsilon_t)$, $y_t = \mathcal{T}_\theta(x_t, \epsilon_t)$, an unbiased estimate of the exact gradient $\nabla_\theta \mathcal{F}_\theta(x_t)$ can be obtained by

$$\nabla_\theta \mathcal{F}_\theta(x_t, \epsilon_t) = \nabla_\theta \min \left\{ 0, \log \frac{\pi(\mathcal{T}_\theta(x_t, \epsilon_t))}{\pi(x_t)} + \log \frac{q_\theta(x_t|\mathcal{T}_\theta(x_t, \epsilon_t))}{q_\theta(\mathcal{T}_\theta(x_t, \epsilon_t)|x_t)} \right\} + \beta \nabla_\theta \mathcal{H}_{q_\theta(y|x_t)},$$

---

**Algorithm 1** Gradient-based Adaptive MCMC

---

**Input:** target $\pi(x)$; reparametrisable proposal $q_\theta(y|x)$ s.t. $y = \mathcal{T}_\theta(x, \epsilon)$, $\epsilon \sim p(\epsilon)$; initial $x_0$; desired average acceptance probability $\alpha_*$.
Initialise $\theta$, $\beta = 1$.
**for** $t = 1, 2, 3, \ldots,$ **do**
  : Propose $\epsilon_t \sim p(\epsilon_t)$, $y_t = \mathcal{T}_\theta(x_t, \epsilon_t)$.
  : Adapt $\theta$: $\theta \leftarrow \theta + \rho_t \nabla_\theta \mathcal{F}_\theta(x_t, \epsilon_t)$.
  : Accept or reject $y_t$ using the standard M-H ratio to obtain $x_{t+1}$.
  : Set $\alpha_t = 1$ if $y_t$ was accepted and $\alpha_t = 0$ otherwise.
  : Adapt hyperparameter $\beta$: $\beta \leftarrow \beta[1 + \rho_\beta(\alpha_t - \alpha_*)]$ # default value for $\rho_\beta = 0.02$.
**end for**

---

which is used to make a gradient update for the parameters $\theta$. Note that the first term in the stochastic gradient is analogous to differentiating through a rectified linear hidden unit (ReLu) in neural networks, i.e. if $\log \frac{\pi(y_t)}{\pi(x_t)} + \log \frac{q_\theta(x_t|y_t)}{q_\theta(y_t|x_t)} \geq 0$ the gradient is zero (this is the case when $y_t$ is accepted with probability one), while otherwise the gradient of the first term reduces to

$$\nabla_\theta \log \pi(\mathcal{T}_\theta(x_t, \epsilon_t)) + \nabla_\theta \log \frac{q_\theta(x_t|\mathcal{T}_\theta(x_t, \epsilon_t))}{q_\theta(\mathcal{T}_\theta(x_t, \epsilon_t)|x_t)}.$$

The value of the hyperparameter $\beta$ can allow to trade off between large acceptance probability and large entropy of the proposal distribution. Such hyperparameter cannot be optimised by maximising the variational objective $\mathcal{F}_\theta$ (this typically will set $\beta$ to a very small value so that the acceptance probability becomes close to one but the chain is not moving since the entropy is very low). Thus, $\beta$ needs to be updated in order to control the average acceptance probability of the chain in order to achieve a certain desired value $\alpha_*$. The value of $\alpha_*$ can be determined based on the specific MCMC proposal we are using and by following standard recommendations in the literature, as discussed also in the previous section. For instance, when we use RWM $\alpha_*$ can be set to $1/4$ (see Section 2.2), while for gradient-based MALA (see Section 2.3) $\alpha_*$ can be set to $0.55$.

Pseudocode for the general procedure is given by Algorithm 1. We set the learning rate $\rho_t$ using RMSProp [39]; at each iteration $t$ we set $\rho_t = \eta/(1 + \sqrt{G_t})$, where $\eta$ is the baseline learning rate, and the updates of $G_t$ depend on the gradient estimate $\nabla_\theta \mathcal{F}_\theta(x_t, \epsilon_t)$ as $G_t = 0.9 G_t + 0.1 \left[\nabla_\theta \mathcal{F}_\theta(x_t, \epsilon_t)\right]^2$.

### 2.2 Fitting a full covariance Gaussian random walk proposal

We now specialise to the case the proposal distribution is a random walk Gaussian $q_L(y|x) = \mathcal{N}(y|x, LL^\top)$ where the parameter $L$ is a positive definite lower triangular matrix, i.e. a Cholesky factor. This distribution is reparametrisable since $y \equiv \mathcal{T}_L(x, \epsilon) = x + L\epsilon$, $\epsilon \sim \mathcal{N}(\epsilon|0, I)$. At the $t$-th iteration when the state is $x_t$ the lower bound becomes

$$\mathcal{F}_L(x_t) = \int \mathcal{N}(\epsilon|0, I) \min\{0, \log \pi(x_t + L\epsilon) - \log \pi(x_t)\} d\epsilon + \beta \sum_{i=1}^{n} \log L_{ii} + \text{const.} \quad (7)$$

Here, the proposal distribution has cancelled out from the M-H ratio, since it is symmetric, while $\mathcal{H}_{q_\theta(y|x_t)} = \log |L| + \text{const}$ and $\log |L| = \sum_{i=1}^{n} \log L_{ii}$. By making use of the MCMC proposed state $y_t = x_t + L\epsilon_t$ we can obtain an unbiased estimate of the exact gradient $\nabla_L \mathcal{F}_L(x_t)$,

$$\nabla_L \mathcal{F}_L(x_t, \epsilon_t) = \begin{cases} \left[\nabla_{y_t} \log \pi(y_t) \times \epsilon_t^\top\right]_{lower} + \beta \text{diag}(\frac{1}{L_{11}}, \ldots, \frac{1}{L_{nn}}), & \text{if } \log \pi(y_t) < \log \pi(x_t) \\ \beta \text{diag}(\frac{1}{L_{11}}, \ldots, \frac{1}{L_{nn}}), & \text{otherwise} \end{cases}$$

where $y_t = x_t + L\epsilon_t$, the operation $[A]_{lower}$ zeros the upper triangular part (above the main diagonal) of a squared matrix and $\text{diag}(\frac{1}{L_{11}}, \ldots, \frac{1}{L_{nn}})$ is a diagonal matrix with elements $1/L_{ii}$. The first case of this gradient, i.e. when $\log \pi(y_t) < \log \pi(x_t)$, has a very similar structure with the stochastic reparametrisation gradient when fitting a variational Gaussian approximation [19, 29, 40] with the difference that here we centre the corresponding approximation, i.e. the proposal $q_L(y_t|x_t)$, at the current state $x_t$ instead of having a global variational mean parameter. Interestingly, this first case when MCMC rejects many samples (or even it gets stuck at the same value for long time) is when

learning can be faster since the term $\nabla_{y_t} \log \pi(y_t) \times \epsilon_t^\top$ transfers information about the curvature of the target to the covariance of the proposal. When we start getting high acceptance rates the second case, i.e. $\log \pi(y_t) \geq \log \pi(x_t)$, will often be activated so that the gradient will often reduce to only having the term $\beta \mathrm{diag}(\frac{1}{L_{11}}, \ldots, \frac{1}{L_{nn}})$ that encourages the entropy of the proposal to become large. The ability to learn from rejections is in sharp contrast with the traditional non gradient-based adaptive MCMC methods which can become very slow when MCMC has high rejection rates. This is because these methods typically learn from the history of state vectors $x_t$ by ignoring the information from the rejected states. The algorithm for learning the full random walk Gaussian follows precisely the general structure of Algorithm 1. For the average acceptance rate $\alpha_*$ we use the value $1/4$.

### 2.3 Fitting a full covariance MALA proposal

Here, we specialise to a full covariance, also called preconditioned, MALA of the form $q_L(y|x) = \mathcal{N}(y|x + (1/2)LL^\top \nabla_x \log \pi(x), LL^\top)$ where the covariance matrix is parametrised by the Cholesky factor $L$. Again this distribution is reparametrisable according to $y \equiv \mathcal{T}_L(x, \epsilon) = x + (1/2)LL^\top \nabla \log \pi(x) + L\epsilon$, $\epsilon \sim \mathcal{N}(\epsilon|0, I)$. At the $t$-th iteration when the state is $x_t$ the reparametrised lower bound simplifies significantly and reduces to,

$$
\mathcal{F}_L(x_t) = \int \mathcal{N}(\epsilon|0, I) \min \Big\{ 0, \log \pi \Big( x_t + (1/2)LL^\top \nabla \log \pi(x_t) + L\epsilon \Big) - \log \pi(x_t)
$$

$$
- \frac{1}{2} \Big( ||(1/2)L^\top [\nabla \log \pi(x_t) + \nabla \log \pi(y)] + \epsilon||^2 - ||\epsilon||^2 \Big) \Big\} d\epsilon + \beta \sum_{i=1}^{n} \log L_{ii} + \text{const},
$$

where $||\cdot||$ denotes Euclidean norm and in the term $\nabla \log \pi(y)$, $y = x_t + (1/2)LL^\top \nabla \log \pi(x_t) + L\epsilon$. Then, based on the proposed state $y_t = \mathcal{T}_L(x_t, \epsilon_t)$ we can obtain the unbiased gradient estimate $\nabla \mathcal{F}_L(x_t, \epsilon_t)$ similarly to the previous section. In general, such an estimate can be very expensive because the existence of $L$ inside $\nabla \log \pi(y_t)$ means that we need to compute the matrix of second derivatives or Hessian $\nabla \nabla \log \pi(y_t)$. We have found that an alternative procedure which stops the gradient inside $\nabla \log \pi(y_t)$ (i.e. it views $\nabla \log \pi(y_t)$ as a constant w.r.t. $L$) has small bias and works well in practice. In fact, as we will show in the experiments this approximation not only is computationally much faster but remarkably also it leads to better adaptation compared to the exact Hessian procedure, presumably because by not accounting for the gradient inside $\nabla \log \pi(y_t)$ reduces the variance. Furthermore, the expression of the gradient w.r.t. $L$ used by this fast approximation can be computed very efficiently with a single $O(n^2)$ operation (an outer vector product; see Supplement), while each iteration of the algorithm requires overall at most four $O(n^2)$ operations. For these gradient-based adaptive MALA schemes, $\beta$ in Algorithm 1 is adapted to obtain an average acceptance rate roughly $\alpha_* = 0.55$.

## 3 Related Work

Connection of our method with traditional adaptive MCMC methods has been discussed in Section 1. Here, we analyse additional related works that make use of gradient-based optimisation and specialised objective functions or algorithms to train MCMC proposal distributions.

The work in [21] proposed a criterion to tune MCMC proposals based on maximising a modified version of the expected squared jumped distance, $\int q_\theta(y|x_t)||y - x_t||^2 \alpha(x_t, y; \theta) dy$, previously considered in [27]. Specifically, the authors in [21] firstly observe that the expected squared jumped distance may not encourage mixing across all dimensions of $x$[1] and then try to resolve this by including a reciprocal term (see Section 4.2 in their paper). The generalised speed measure proposed in this paper is rather different from such criteria since it encourages joint exploration of all dimensions of $x$ by applying maximum entropy regularisation, which by construction penalises "dimensions that do not move" since the entropy becomes minus infinity in such cases. Another important difference is that in our method the optimisation is performed in the log space by propagating gradients through the logarithm of the M-H acceptance probability, i.e. through $\log \alpha(x_t, y; \theta)$ and not through $\alpha(x_t, y; \theta)$. This is exactly analogous to other numerically stable objectives such as variational lower bounds and log likelihoods, and as those our method leads to numerically stable optimisation for arbitrarily large dimensionality of $x$ and complex targets $\pi(x)$.

In another related work, the authors in [25] considered minimising the KL divergence $\mathrm{KL}[\pi(x_t)q_\theta(y_t|x_t)||\pi(y_t)q_\theta(x_t|y_t)]$. However, this loss for standard proposal schemes, such as RWM and MALA, leads to degenerate deterministic solutions where $q_\theta(y_t|x_t)$ collapses to a delta function. Therefore, [25] maximised this objective for the independent M-H sampler where the collapsing problem does not occur. The entropy regularised objective we introduced is different and it can adapt arbitrary MCMC proposal distributions, and not just the independent M-H sampler.

There has been also work to learn flexible MCMC proposals using neural networks [38, 21, 16, 36]. For instance, [38] use volume preserving flows and an adversarial objective, [21] use the modified expected jumped distance, discussed earlier, to learn neural network-based extensions of HMC, while [16, 36] use auxiliary variational inference. The need to train neural networks can add a significant computational cost, and from the end-user point of view these neural adaptive samplers might be hard to tune especially in high dimensions. Notice that the generalised speed measure we proposed in this paper could possibly be used to train neural adaptive samplers as well. However, to really obtain practical algorithms we need to ensure that training has small cost that does not overwhelm the possible benefits in terms of effective sample size.

Finally, the generalised speed measure that is based on entropy regularisation shares similarities with other used objectives for learning probability distributions, such as in variational Bayesian inference, where the variational lower bound includes an entropy term [18, 10] and reinforcement learning (RL) where maximum-entropy regularised policy gradients are able to estimate more explorative policies [37, 23]. Further discussion on the resemblance of our algorithm with RL is given in the Supplement.

## 4   Experiments

We test the gradient-based adaptive MCMC methods in several simulated and real data. We investigate the performance of two instances of the framework: the gradient-based adaptive random walk (gadRWM) detailed in Section 2.2 and the corresponding MALA (gadMALA) detailed in Section 2.3. For gadMALA we consider the exact reparametrisation method that requires the evaluation of the Hessian (gadMALAe) and the fast approximate variant (gadMALAf). These schemes are compared against: (i) standard random walk Metropolis (RWM) with proposal $\mathcal{N}(y|x, \sigma^2 I)$, (ii) an adaptive MCMC (AM) that fits a proposal of the form $\mathcal{N}(y|x, \Sigma)$ (we consider a computational efficient version based on updating the Cholesky factor of $\Sigma$; see Supplement), (iii) a standard MALA proposal $\mathcal{N}(y|x + (1/2)\sigma^2 \nabla \log \pi(x), \sigma^2 I)$, (iv) an Hamiltonian Monte Carlo (HMC) with a fixed number of leap frog steps (either 5, or 10, or 20) (v) and the state of the art no-U-turn sampler (NUTS) [17] which arguably is the most efficient adaptive HMC method that automatically determines the number of leap frog steps. We provide our own MALTAB implementation[2] of all algorithms, apart from NUTS for which we consider a publicly available implementation.

### 4.1   Illustrative experiments

To visually illustrate the gradient-based adaptive samplers we consider a correlated 2-D Gaussian target with covariance matrix $\Sigma = [1\ 0.99; 0.99\ 1]$ and a 51-dimensional Gaussian target obtained by evaluating the squared exponential kernel plus small white noise, i.e. $k(x_i, x_j) = \exp\{-\frac{1}{2}\frac{(x_i - x_j)^2}{0.16}\} + 0.01\delta_{i,j}$, on the regular grid $[0, 4]$. The first two panels in Figure 1 show the true covariance together with the adapted covariances obtained by gadRWM for two different settings of the average acceptance rate $\alpha_*$ in Algorithm 1, which illustrates also the adaptation of the entropy-regularisation hyperparameter $\beta$ that is learnt to obtain a certain $\alpha_*$. The remaining two plots illustrate the ability to learn a highly correlated 51-dimensional covariance matrix (with eigenvalues ranging from 0.01 to 12.07) by applying our most advanced gadMALAf scheme.

### 4.2   Quantitative results

Here, we compare all algorithms in some standard benchmark problems, such as Bayesian logistic regression, and report effective sample size (ESS) together with other quantitative scores.

**Experimental settings.** In all experiments for AM and gradient-based adaptive schemes the Cholesky factor $L$ was initialised to a diagonal matrix with values $0.1/\sqrt{n}$ in the diagonal where $n$ is the dimensionality of $x$. For the AM algorithm the learning rate was set to $0.001/(1 + t/T)$ where $t$ is the number of iterations and $T$ (the value 4000 was used in all experiments) serves to keep the learning rate nearly constant for the first $T$ training iterations. For the gradient-based adaptive algorithms

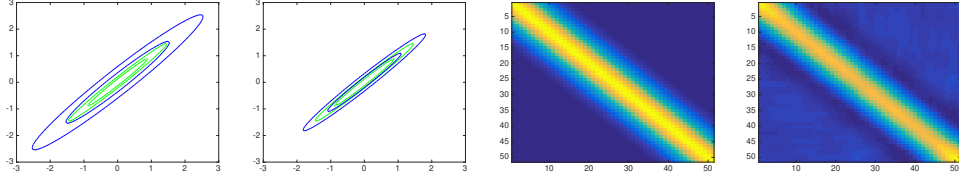

**Figure 1:** The green contours in the first two panels (from left to right) show the 2-D Gaussian target, while the blue contours show the learned covariance, $LL^\top$, after adapting for $2 \times 10^4$ iterations using gadRWM and targeting acceptance rates $\alpha_* = 0.25$ and $\alpha_* = 0.4$, respectively. For $\alpha_* = 0.25$ the adapted blue contours show that the proposal matches the shape of the target but it has higher entropy/variance and the hyperparameter $\beta$ obtained the value 7.4. For $\alpha_* = 0.4$ the blue contours shrink a bit and $\beta$ is reduced to 2.2 (since higher acceptance rate requires smaller entropy). The third panel shows the exact $51 \times 51$ covariance matrix and the last panel shows the adapted one, after running our most efficient gadMALAf scheme for $2 \times 10^5$ iterations. In both experiments $L$ was initialised to diagonal matrix with $0.1/\sqrt{n}$ in the diagonal.

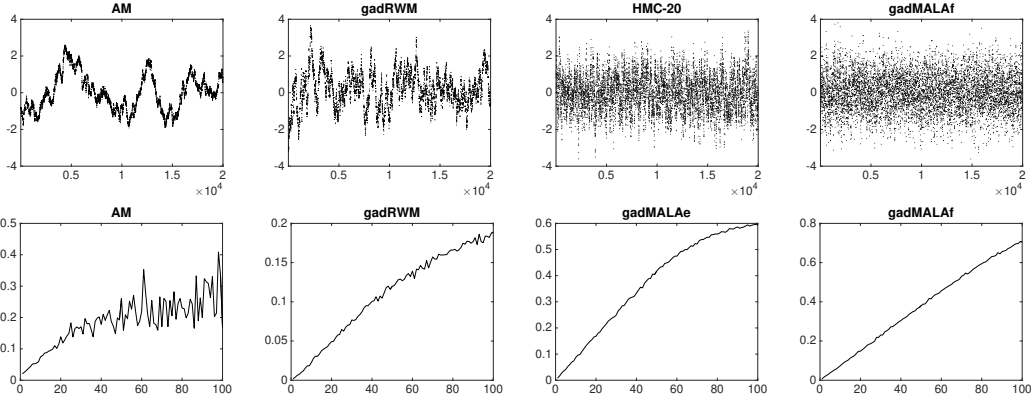

**Figure 2:** Panels in the first row show trace plots, obtained by different schemes, across the last $2 \times 10^4$ sampling iterations for the most difficult to sample $x_{100}$ dimension. The panels in the second row show the estimated values of the diagonal of $L$ obtained by different adaptive schemes. Notice that the real Gaussian target has diagonal covariance matrix $\Sigma = \text{diag}(s_1^2, \ldots, s_{100}^2)$ where $s_i$ are uniform in the range $[0.01, 1]$.

we use RMSprop (see Section 2.1) where $\eta$ was set to $0.00005$ for gadRWM and to $0.00015$ for the gadMALA schemes. NUTS uses its own fully automatic adaptive procedure that determines both the step size and the number of leap frog steps [17]. For all experiments and algorithms (apart from NUTS) we consider $2 \times 10^4$ burn-in iterations and $2 \times 10^4$ iterations for collecting samples. This adaptation of $L$ or $\sigma^2$ takes place only during the burn-in iterations and then it stops, i.e. at collection of samples stage these parameters are kept fixed. For NUTS, which has its own internal tuning procedure, 500 burn-in iterations are sufficient before collecting $2 \times 10^4$ samples. The computational time for all algorithms reported in the tables corresponds to the overall running time, i.e. the time for performing jointly all burn-in and collection of samples iterations.

**Neal's Gaussian target.** We first consider an example used in [24] where the target is a zero-mean multivariate Gaussian with diagonal covariance matrix $\Sigma = \text{diag}(s_1^2, \ldots, s_{100}^2)$ where the stds $s_i$ take values in the linear grid $0.01, 0.02, \ldots, 1$. This is a challenging example because the different scaling of the dimensions means that the schemes that use an isotropic step $\sigma^2$ will be adapted to the smallest dimension $x_1$ while the chain at the higher dimensions, such as $x_{100}$, will be moving slowly exhibiting high autocorrelation and small effective sample size. The first row of Figure 3 shows the trace plot across iterations of the dimension $x_{100}$ for some of the adaptive schemes including an HMC scheme that uses 20 leap frog steps. Clearly, the gradient-based adaptive methods show much smaller autocorrelation that AM. This is because they achieve a more efficient adaptation of the Cholesky factor $L$ which ideally should become proportional to a diagonal matrix with the linear grid $0.01, 0.02, \ldots, 1$ in the main diagonal. The second row of Figure 3 shows the diagonal elements of $L$ from which we can observe that all gradient-based schemes lead to more accurate adaptation with gadMALAf being the most accurate.

Furthermore, Table 1 provides quantitative results such as minimum, median and maximum ESS computed across all dimensions of the state vector $x$, running times and an overall efficiency score

**Table 1:** Comparison in Neal's Gaussian example (dimensionality was $n = 100$; see panel above) and Caravan binary classification dataset where the latter consists of 5822 data points (dimensionality was $n = 87$; see panel below). All numbers are averages across ten repeats where also one-standard deviation is given for the Min ESS/s score. From the three HMC schemes we report only the best one in each case.

| Method | Time(s) | Accept Rate | ESS (Min, Med, Max) | Min ESS/s (1 st.d.) |
|---|---|---|---|---|
| *(Neal's Gaussian)* | | | | |
| gadMALAf | 8.7 | 0.556 | (1413.4, 1987.4, 2580.8) | **161.70** (15.07) |
| gadMALAe | 10.0 | 0.541 | (922.2, 2006.3, 2691.1) | 92.34 (7.11) |
| gadRWM | 7.0 | 0.254 | (27.5, 66.9, 126.9) | 3.95 (0.66) |
| AM | 2.3 | 0.257 | (8.7, 48.6, 829.1) | 3.71 (0.87) |
| RWM | 2.2 | 0.261 | (2.9, 8.4, 2547.6) | 1.31 (0.06) |
| MALA | 3.1 | 0.530 | (2.9, 10.0, 12489.2) | 0.95 (0.03) |
| HMC-20 | 49.7 | 0.694 | (306.1, 1537.8, 19732.4) | 6.17 (3.35) |
| NUTS | 360.5 | >0.7 | (18479.6, 20000.0, 20000.0) | 51.28 (1.64) |
| *(Caravan)* | | | | |
| gadMALAf | 23.1 | 0.621 | (228.1, 750.3, 1114.7) | **9.94** (2.64) |
| gadMALAe | 95.1 | 0.494 | (66.6, 508.3, 1442.7) | 0.70 (0.16) |
| gadRWM | 22.6 | 0.234 | (5.3, 34.3, 104.5) | 0.23 (0.06) |
| AM | 20.0 | 0.257 | (3.2, 11.8, 62.5) | 0.16 (0.01) |
| RWM | 15.3 | 0.242 | (3.0, 9.3, 52.5) | 0.20 (0.03) |
| MALA | 22.8 | 0.543 | (4.4, 28.3, 326.0) | 0.19 (0.05) |
| HMC-10 | 225.5 | 0.711 | (248.3, 2415.7, 19778.7) | 1.10 (0.12) |
| NUTS | 1412.1 | >0.7 | (7469.5, 20000.0, 20000.0) | 5.29 (0.38) |

Min ESS/s (i.e. ESS for the slowest moving component of $x$ divided by running time – last column in the Table) which allows to rank the different algorithms. All results are averages after repeating the simulations 10 times under different random initialisations. From the table it is clear that the gadMALA algorithms give the best performance with gadMALAf being overall the most effective.

**Bayesian logistic regression.** We consider binary classification where given a set of training examples $\{y_i, s_i\}_{i=1}^n$ we assume a logistic regression likelihood $p(y|w, s) = \sum_{i=1}^n y_i \log \sigma(s_i) + (1 - y_i) \log(1 - \sigma(s_i))$, where $\sigma(s_i) = 1/(1 + \exp(-w^\top s_i))$, $s_i$ is the input vector and $w$ the parameters. We place a Gaussian prior on $w$ and we wish to sample from the posterior distribution over $w$. We considered six binary classification datasets (Australian Credit, Heart, Pima Indian, Ripley, German Credit and Caravan) with a number of examples ranging from $n = 250$ to $n = 5822$ and dimensionality of the state/parameter vector ranging from 3 to 87. Table 1 shows results for the most challenging Caravan dataset where the dimensionality of $w$ is 87, while the remaining five tables are given in the Supplement. From all tables we observe that the gadMALAf is the most effective and it outperforms all other methods. While NUTS has always very high ESS is still outperformed by gadMALAf because of the high computational cost, i.e. NUTS might need to use a very large number of leap frog steps (each requiring re-evaluating the gradient of the log target) per iteration. Further results, including a higher 785-dimensional example on MNIST, are given in the Supplement.

## 5 Conclusion

We have presented a new framework for gradient-based adaptive MCMC that makes use of an entropy-regularised objective function that generalises the concept of speed measure. We have applied this method for learning RWM and MALA proposals with full covariance matrices.

Some topics for future research are the following. Firstly, to deal with very high dimensional spaces it would be useful to consider low rank parametrisations of the covariance matrices in RWM and MALA proposals. Secondly, it would be interesting to investigate whether our method can be used to tune the so-called mass matrix in HMC samplers. However, in order for this to lead to practical and scalable algorithms we have to come up with schemes that avoid the computation of the Hessian, as we successfully have done for MALA. Finally, in order to reduce the variance of the stochastic gradients and speed up further the adaptation, especially in high dimensions, our framework could be possibly combined with parallel computing as used for instance in deep reinforcement learning [12].

## Footnotes

[1]Because the additive form of $||y - x_t||^2 = \sum_i (y_i - x_{ti})^2$ implies that even when some dimensions might not be moving at all (the corresponding distance terms are zero), the overall sum can still be large.

[2]https://github.com/mtitsias/gadMCMC.

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
