[Supplementary Material]

# Supplement:
# Gradient-based Adaptive Markov Chain Monte Carlo

## A   Gradient-based adaptive MCMC as Reinforcement Learning

A MDP is a tuple $(\mathcal{X}, \mathcal{Y}, \mathcal{P}, \mathcal{R})$ where $\mathcal{X}$ is the state space, $\mathcal{Y}$ is the action space, $\mathcal{P}$ is the transition distribution with density $p(x_{t+1}|x_t, y_t)$ that describes how the next state $x_{t+1}$ is generated given that currently we are at state $x_t$ and we take action $y_t$. Further, the reward function $R(x_t, y_t)$ provides some instantaneous or local signal about how good the action $y_t$ was when being at $x_t$. Furthermore, in a MDP we have also a policy $\pi(y_t|x_t)$ which is a distribution over actions given states and it fully describes the behaviour of the agent. Given that we start at $x_0$ we wish to specify the policy so that to maximise future reward, such as the expected accumulated discounted reward

$$\mathbb{E}_{\pi}\left[\sum_{t=0}^{\infty}\gamma^t R(x_t, y_t)\right], \quad \gamma \in [0, 1].$$

Suppose now a MCMC procedure targeting $\pi(x)$, where $x \in \mathcal{X}$ is the state vector. Consider a proposal distribution $q_\theta(y|x)$, such that the standard Metropolis-Hastings algorithm accepts each proposed state $y_t \sim q_\theta(y_t|x_t)$ with probability

$$\alpha(x_t, y_t; \theta) = \min\left(1, \frac{\pi(y_t)}{\pi(x_t)}\frac{q_\theta(x_t|y_t)}{q_\theta(y_t|x_t)}\right), \tag{8}$$

so that $x_{t+1} = y_t$, while if the proposal is rejected, $x_{t+1} = x_t$. To reformulate MCMC as an MDP we make the following correspondences. Firstly, both the state $x_t$ and the action $y_t$ will live in the same space which will be the state space $\mathcal{X}$ of the target distribution. The MCMC proposal $q_\theta(y_t|x_t)$ will correspond to the policy $\pi(y_t|x_t)$, while the environmental transition dynamics will be stochastic and given by the two-component mixture,

$$p(x_{t+1}|x_t, y_t) = \alpha(x_t, y_t; \theta)\delta_{x_{t+1}, y_t} + (1 - \alpha(x_t, y_t; \theta))\delta_{x_{t+1}, x_t},$$

where $\delta_{x,y}$ denotes the delta function. This transition density simply says that the new state $x_{t+1}$ with probability $\alpha(x_t, y_t; \theta)$ will be equal to the proposed action $y_t$, while with the remaining probability will be set to the previous state, i.e. $x_{t+1} = x_t$. Notice that the standard MCMC transition kernel $K_\theta(x_t, x_{t+1})$ is obtained by integrating out the action $y_t$, i.e.

$$K_\theta(x_t, x_{t+1}) = \int p(x_{t+1}|x_t, y_t)q_\theta(y_t|x_t)dy_t$$

$$= \alpha(x_t, x_{t+1})q(x_{t+1}|x_t) + \left(1 - \int \alpha(x_t, y_t)q_\theta(y_t|x_t)dy_t\right)\delta_{x_{t+1}, x_t}. \tag{9}$$

The final ingredient we need to reformulate MCMC as MDP is the reward function $R(x_t, y_t)$. The gradient-based adaptive MCMC method essentially assumes as reward

$$R(y_t, x_t; \theta) = \log \alpha(x_t, y_t; \theta) - \beta \log q_\theta(y_t|x_t),$$

which is an entropy-regularised reward that promotes high exploration with the entropic term $-\beta \log q_\theta(y_t|x_t)$. Gradient-based adaptive MCMC essentially at each step stochastically maximises the expected reward starting from state $x_t$, i.e.

$$\int q_\theta(y_t|x_t)R(y_t, x_t; \theta)dy_t.$$

While the above reformulates MCMC as a reinforcement learning (RL) problem, there are clearly also some differences with standard RL problems. Given that the reward $R(y_t, x_t; \theta)$ is very informative (we are not facing the delayed-reward problem commonly encountered in standard RL) gradient-based MCMC sets $\gamma = 0$ in order to maximise immediate reward. Further, the transition dynamics $p(x_{t+1}|x_t, y_t)$ are known in MCMC, while this typically is not the case in standard RL. Finally, notice that the reward $R(y_t, x_t; \theta)$ as well as the transition dynamics $p(x_{t+1}|x_t, y_t)$ all depend on the parameter $\theta$ and the policy $q_\theta(y_t|x_t)$, i.e. they depend on the MCMC proposal distribution.

# B  Further details about the algorithms

For the standard adaptive MCMC method (AM) we implemented a computational efficient version that requires no matrix decompositions (which are expensive due to the $O(n^3)$ scaling) by parametrising the proposal as $\mathcal{N}(y|x, LL^\top)$ and updating the Cholesky factor in each iteration according to the updates

$$\mu \leftarrow \mu + \rho_t(x_{t+1} - \mu),$$

$$L \leftarrow L + \rho_t L \left[ L^{-1}(x_{t+1} - \mu)(x_{t+1} - \mu)^\top L^{-\top} - I \right]_{lower},$$

where $\mu$ tracks the global mean of the state vector. Further details about this scheme can be found in Section 5.1.1 in [3].

For our most efficient gadMALAf scheme the stochastic gradient in each iteration is

$$\nabla \mathcal{F}_L(x_t, \epsilon_t) = \nabla_L \min \Big\{ 0, \log \pi \left( x_t + (1/2) LL^\top \nabla \log \pi(x_t) + L\epsilon_t \right) - \log \pi(x_t)$$

$$- \frac{1}{2} \left( ||(1/2)L^\top[\nabla \log \pi(x_t) + \nabla \log \pi(y_t)] + \epsilon_t||^2 - ||\epsilon_t||^2 \right) \Big\} + \beta \nabla_L \sum_{i=1}^n \log L_{ii},$$

where, as discussed in the main paper, $\nabla \log \pi(y_t)$ is taken as constant w.r.t. $L$. Then the gradient of the M-H log ratio (when this log ratio is negative, since otherwise its gradient is zero) simplifies as

$$\nabla_L \log \pi \left( x_t + (1/2) LL^\top \nabla \log \pi(x_t) + L\epsilon_t \right) - \frac{1}{2} \nabla_L ||(1/2)L^\top[\nabla \log \pi(x_t) + \nabla \log \pi(y_t)] + \epsilon_t||^2$$

$$= -\frac{1}{2} \left( \nabla \log \pi(x_t) - \nabla \log \pi(y_t) \right) \left( (1/2)L^\top[\nabla \log \pi(x_t) - \nabla \log \pi(y_t)] + \epsilon_t \right)^\top$$

and then take the lower triangular part. This is just an outer vector product that scales as $O(n^2)$. Overall each iteration of the algorithm can be implemented (plus the extra overhead of a single gradient evaluation $\nabla \log \pi(y_t)$ of the log target at the proposed state $y_t$) by using at most four $O(n^2)$ operations during adaptation and exactly two $O(n^2)$ operations after burn-in, as shown in the released code.

# C  Extra results on the Neal's Gaussian target

Figure 3 shows trace plot for the log density of the target for all different algorithms.

**Figure 3:** The evolution of the log-target across iterations for all algorithms in Neal's Gaussian example.

**Table 2:** Comparison of sampling methods in Australian Credit dataset consisted of 690 data points. The size of the state/parameter vector from which we draw samples was $n = 15$.

| Method | Time(s) | Accept Rate | ESS (Min, Med, Max) | Min ESS/s (1 st.d.) |
|---|---|---|---|---|
| gadMALAf | 8.1 | 0.569 | (3485.9, 4262.9, 4784.0) | 443.97 (76.13) |
| gadMALAe | 13.5 | 0.540 | (3034.9, 4234.3, 4836.3) | 227.61 (29.87) |
| gadRWM | 7.7 | 0.253 | (288.0, 423.0, 515.0) | 38.68 (9.53) |
| AM | 4.4 | 0.261 | (310.9, 410.1, 507.2) | 70.21 (6.23) |
| RWM | 3.4 | 0.252 | (31.3, 312.6, 495.2) | 9.16 (3.12) |
| MALA | 7.0 | 0.524 | (138.4, 2388.1, 3818.8) | 20.22 (5.14) |
| HMC-5 | 37.0 | 0.700 | (1048.1, 3510.3, 14809.7) | 28.06 (11.69) |
| NUTS | 41.3 | >0.7 | (2995.2, 20000.0, 20000.0) | 72.86 (7.31) |

# D   Extra results on the binary classification datasets

Tables 2-6 show the results for the remaining five binary classification datasets not reported in the main article. Figures 4 and 5 show trace plot of log density of the target acrorss all different algorithms for the German Credit and Caravan datasets. For the remaining datasets the corresponding plots are similar.

**Table 3:** Comparison of sampling methods in Ripley dataset consisted of 250 data points. The size of the state/parameter vector from which we draw samples was $n = 3$.

| Method | Time(s) | Accept Rate | ESS (Min, Med, Max) | Min ESS/s (1 st.d.) |
|--------|---------|-------------|---------------------|---------------------|
| gadMALAf | 3.3 | 0.536 | (8328.4, 8913.2, 9442.4) | 2506.04 (143.47) |
| gadMALAe | 4.9 | 0.543 | (8446.7, 9006.6, 9595.6) | 1713.44 (44.91) |
| gadRWM | 3.1 | 0.068 | (638.0, 736.9, 803.2) | 205.99 (17.85) |
| AM | 3.0 | 0.257 | (1702.8, 1792.2, 1902.0) | 570.19 (49.32) |
| RWM | 2.1 | 0.252 | (1129.2, 1627.8, 1979.8) | 534.21 (43.54) |
| MALA | 2.8 | 0.542 | (2976.0, 5683.0, 9726.5) | 1046.48 (54.86) |
| HMC-5 | 14.7 | 0.678 | (9205.3, 10818.1, 16136.5) | 626.55 (196.48) |
| NUTS | 7.5 | >0.7 | (9436.2, 17463.5, 20000.0) | 1265.99 (73.01) |

**Table 4:** Comparison of sampling methods in Pima Indian dataset consisted of 532 data points. The size of the state/parameter vector from which we draw samples was $n = 8$.

| Method | Time(s) | Accept Rate | ESS (Min, Med, Max) | Min ESS/s (1 st.d.) |
|--------|---------|-------------|---------------------|---------------------|
| gadMALAf | 4.6 | 0.545 | (5407.6, 5810.3, 6467.6) | 1176.12 (79.54) |
| gadMALAe | 6.8 | 0.547 | (5469.6, 5963.6, 6421.1) | 801.03 (16.07) |
| gadRWM | 4.2 | 0.267 | (635.6, 760.0, 866.2) | 150.70 (9.73) |
| AM | 4.1 | 0.273 | (612.7, 729.1, 854.8) | 149.18 (10.40) |
| RWM | 3.2 | 0.246 | (354.6, 496.4, 709.6) | 111.81 (6.16) |
| MALA | 4.0 | 0.509 | (1524.9, 2457.2, 3853.6) | 377.17 (25.80) |
| HMC-5 | 20.3 | 0.711 | (7295.7, 12798.7, 18267.4) | 359.22 (103.55) |
| NUTS | 15.2 | >0.7 | (15343.3, 18606.0, 20000.0) | 1008.97 (42.33) |

**Table 5:** Comparison of sampling methods in Heart dataset consisted of 270 data points. The size of the state/parameter vector from which we draw samples was $n = 14$.

| Method | Time(s) | Accept Rate | ESS (Min, Med, Max) | Min ESS/s (1 st.d.) |
|--------|---------|-------------|---------------------|---------------------|
| gadMALAf | 4.1 | 0.551 | (3892.9, 4362.7, 4784.2) | 946.98 (56.10) |
| gadMALAe | 6.4 | 0.560 | (3832.4, 4372.3, 4845.6) | 599.51 (30.00) |
| gadRWM | 3.8 | 0.288 | (342.5, 440.9, 536.1) | 88.94 (10.29) |
| AM | 3.2 | 0.238 | (342.5, 425.5, 535.4) | 106.97 (7.18) |
| RWM | 2.3 | 0.266 | (196.9, 314.3, 472.7) | 86.57 (11.33) |
| MALA | 3.5 | 0.530 | (1429.7, 2310.6, 3260.4) | 404.96 (18.57) |
| HMC-5 | 18.4 | 0.699 | (1913.2, 5600.3, 11883.0) | 103.81 (39.38) |
| NUTS | 15.4 | >0.7 | (20000.0, 20000.0, 20000.0) | 1295.13 (15.74) |

**Table 6:** Comparison of sampling methods in German Credit dataset consisted of 1000 data points. The size of the state/parameter vector from which we draw samples was $n = 25$.

| Method | Time(s) | Accept Rate | ESS (Min, Med, Max) | Min ESS/s (1 st.d.) |
|--------|---------|-------------|---------------------|---------------------|
| gadMALAf | 11.0 | 0.560 | (2734.9, 3414.5, 3928.6) | 252.91 (37.86) |
| gadMALAe | 22.4 | 0.549 | (2808.2, 3384.9, 3883.5) | 126.00 (14.68) |
| gadRWM | 10.4 | 0.248 | (179.1, 252.5, 323.1) | 17.92 (4.18) |
| AM | 12.6 | 0.262 | (121.9, 207.6, 308.0) | 9.72 (1.25) |
| RWM | 8.4 | 0.233 | (45.0, 153.8, 298.7) | 5.48 (1.83) |
| MALA | 9.2 | 0.535 | (420.1, 1313.2, 2573.7) | 47.37 (10.22) |
| HMC-5 | 43.4 | 0.706 | (3020.2, 10294.4, 20000.0) | 71.62 (49.62) |
| NUTS | 47.4 | >0.7 | (7737.2, 20000.0, 20000.0) | 166.93 (30.57) |

**Figure 4:** The evolution of the log-target across iterations for all algorithms in German Credit dataset.

## E    Results on a higher dimensional example

We all tried a much larger Bayesian binary classification problem by taking all $11339$ training examples of "5" and "6" MNIST digits which are $28 \times 28$ images and therefore the dimensionality of the parameter vector $w$ was 785 (the plus one accounts for the bias term). For this larger example from the baselines we applied the gradient-based schemes, MALA, HMC and NUTS since the other methods become very inefficient. From the proposed schemes we applied the most efficient algorithm which is gadMALAf. Also because of the much higher dimensionality of this problem, which makes the stochastic optimisation over $L$ harder, we had to decrease the baseline learning rate in the RMSprop schedule from $0.00015$ to $0.00001$. We also considered a larger adaptation phase consisted of $5 \times 10^4$ instead of $2 \times 10^4$. All other algorithms use the same experimental settings as described in the main paper. Figure 7 shows the evolution of the log-target densities for all sampling schemes while Table 7 gives ESS, computation times and other statistics.

We can observe that the performance of gadMALAf is reasonably good: it outperforms all methods apart form NUTS. NUTS is better in this example, but it takes around 22 hours to run (since it performs on average 550 gradient evaluations per sampling iteration). Finally, to visualise some part of the learned $L$ found by gadMALAf, Figure 7 depicts the 784 diagonal elements of $L$ as an $28 \times 28$ grey-scale image. Clearly, gadMALAf manages to perform a sort of feature selection, i.e. to discover that the border pixels in MNIST digits do not really take part in the classification, so it learns a much higher variance for those dimensions (close to the variance of the prior).

**Figure 5:** The evolution of the log-target across iterations for all algorithms in Caravan dataset.

**Figure 6:** The evolution of the log-target across iterations for all algorithms in binary MNIST classification over "5" versus "6".

**Table 7:** Comparison of sampling methods in binary MNIST dataset, of "5" versus "6", consisted of 11339 data points. The size of the state/parameter vector from which we draw samples was $n = 785$. All numbers are averages across five repeats where also one-standard deviation is given for the Min ESS/s score.

| Method | Time(s) | Accept Rate | ESS (Min, Med, Max) | Min ESS/s (1 st.d.) |
|---|---|---|---|---|
| gadMALAf | 779.3 | 0.575 | (46.0, 128.7, 282.8) | 0.059 (0.00) |
| MALA | 311.8 | 0.530 | (2.8, 5.9, 28.4) | 0.009 (0.00) |
| HMC-5 | 1847.4 | 0.733 | (4.5, 23.1, 162.7) | 0.002 (0.00) |
| HMC-10 | 3381.3 | 0.589 | (13.9, 66.5, 576.0) | 0.004 (0.00) |
| HMC-20 | 6449.1 | 0.666 | (77.8, 240.1, 2060.9) | 0.012 (0.00) |
| NUTS | 83232.1 | >0.7 | (18514.1, 20000.0, 20000.0) | 0.223 (0.01) |

**Figure 7:** The first 784 diagonal elements (i.e. excluding the bias component of $x$) of the full $785 \times 785$ Cholesky factor $L$ found after $5 \times 10^4$ adapting iterations by gadMALAf. Brighter/white colour means larger values.