[Reviews · NeurIPS 2019]

Reviewer 1



EDIT: After reading the feedback of the other reviewers I realized that I was too harsh in the assessment of the paper. Consequently, I updated my assessment of the paper to a good submission that should be accepted at the conference. The paper in itself is well written, and the explanations are clear and concise. However, I don't feel NeurIPS is the appropriate venue to publish it. My main concern about the paper is with respect to its appeal for the ML community at large. My impression is that its scope is rather limited. The presented examples are of reduced dimension and it is not clear for me how the method could scale to large datasets. There are some leads given in the conclusion section, however I would have preferred to actually see them implemented instead of just being mentioned.

Reviewer 2



Originality: First-order Gradient-based MCMC methods have to deal with determining an appropriate length scale for each variable. NUTS is one approach and this paper gives another approach whereby a parameter theta of a proposal distribution is adaptively improved to account for the covariance structure. At the same time theta is adapted to consider the entropy of the proposal distribution. This trade off for theta is rolled into a new speed measure which is the central point of this paper. The paper includes a lower bound of the speed measure that can be directly differentiated resulting in a practical algorithm. The paper also includes a heuristic that makes this adaptive MCMC algorithm applicable to MALA as well. All of these ideas are very original. Clarity: The paper is quite clear, but maybe the authors could include more results from the supplement into the main text. Quality: The technical quality is good and the results confirm the claim that the new algorithm is definitely more sample efficient than NUTS. This makes this work a very important contribution to the field. There are only a few results in the main paper, but the supplement has a lot more results. Significance: The results are very important because they have provided an adaptive approach that can be applied to even the trivial random walk proposers as well as the more sophisticated Langevin proposers. Given the overall simplicity of the approach this would be a very strong competitor for the popularly used NUTS algorithm, specially in light of the presented results, and would trigger further research. Some minor comments: It was not clear how the Adaptive MCMC (AM) described in the results works i.e. what is the objective function of the adaptation. A reference is made to the supplement, but I didn't quite find it there (lines 226-227). Line 266 and Figure 2 top panel. This should really be the auto-correlation plot. It's hard to interpret auto-correlation from a trace plot. The choice of rho_t on lines 136, 137 is not well motivated. In similar situations a Robbins-Monro sequence is typically used. ** POST REBUTTAL ** Thanks for answering the questions and for promising to update the plots. However, I would strongly advise you to provide results comparing with Stan. It will be very hard to take this as an improvement over the state of the art without a comparison to the state of the art!

Reviewer 3



I think this is a very nice paper, which provides a thoughtful approach to an important problem. I found section 2 impressively clear and easy to follow; this is quite well-written. The algorithms for adapting random walk metropolis and MALA to fit full covariance matrices seem quite nice, and a great alternative to fitting covariance matrices by e.g. moment matching to accepted samples, or estimating from independent pre-runs. I think this has potential to see broad use. I also think the generalization of "speed" has potential to be used to derive adaptive variants of other common proposal types. Some minor comments / questions / complaints: * the one thing I find unsatisfying is the need to fall back on existing results for "optimal" acceptance rates when tuning \beta. For RW and MALA proposals, maybe this is even reasonable (there's some theory about optimal preconditioners, proposal covariances, mass matrices anyway…), but it would be nice to address this more generally, or at least discuss the appropriateness of falling back on these recommendations. * line 140, L is described as a "positive definite lower triangular matrix" … presumably a typo (it is correct later for MALA), but LL' is positive definite, while L is lower triangular * the second row of figure 2 is a bit hard to interpret; maybe something else would be better to show. While this should clearly be a monotonic function, I don't have a sense of the optimal choice of scale when used in the different algorithms, or whether it should be linear with the target density variance, or what, particularly for MALA. * It would be good to release code. The "stop gradient" aspect may induce difficulties in implementation, at least as a reference. * For MALA and NUTS, what sort of preconditioning is done as a baseline? For NUTS as implemented in STAN and PyMC3, I know a standard practice is to run a short pre-run which is used to estimate the mass matrix; this is then used fixed with only the step size adapted online. Is something like this done here? What about for the non-adaptive MALA? ====== I read the rebuttal — thanks for your comments.

[Author Response · NeurIPS 2019]

We are grateful to all the reviewers for their feedback. Below we provide responses to the main comments.

**Reviewer 1:**
"I don't feel NeurIPS is the appropriate venue to publish it. My main concern about the paper is with respect to its appeal for the ML community at large. My impression is that its scope is rather limited. The presented examples are of reduced dimension" Although we respect reviewer's opinion we disagree and we believe that our work is very suitable for NeurIPS which is an interdisciplinary conference and MCMC is one of its subject areas. Many MCMC papers and related Monte Carlo methods have been previously published in NeurIPS. Also not all inference problems in statistics and ML are large scale or high-dimensional and certainly our method does not exclude applicability to high dimensions.

**Reviewer 2:**
"It was not clear how the Adaptive MCMC (AM) described in the results works i.e. what is the objective function of the adaptation. A reference is made to the supplement, but I didn't quite find it there (lines 226-227)." The underlying objective of AM is the minimisation of the KL divergence between the target distribution and the proposal distribution, as described earlier in section 4.1 in the tutorial paper of Andrieu and Thoms. Of course this optimisation is challenging because we only observe correlated samples which at the early adaptation stages are really far from the target.

"Line 266 and Figure 2 top panel. This should really be the auto-correlation plot." Thank you, we will follow your suggestion.

"The choice of $\rho_t$ on lines 136, 137 is not well motivated. In similar situations a Robbins-Monro sequence is typically used." We agree that the Robbins-Monro sequence is the one that ensures convergence in the limit. The motivation behind the RMSprop sequence we used (that is very popular in Deep Learning together with similar adaptive learning rate schemes such as Adam) is that in practise when you run for a fixed (relatively small) budget of stochastic optimisation/adaptation iterations it tends to provide more effective optimisation. Note that in our experiemtns we adapt only during burn-in, while at the collection of samples phase we keep the proposal fixed.

"The authors could give more theoretical justification for their MALA approximation to avoid computing the Hessian on lines 172-174." So far we only empirically observe that the fast MALA scheme tends to provide stostastic gradients with smaller variance leading to faster optimisation. We will try to analyse this theoretically by finding expressions of the variance (at least for simple targets) for the exact Hessian and the fast scheme.

" I would have like to see the results for Stan, which has a good implementation of NUTS, on the presented data sets. That would help highlight the advantage of the proposed method over the current standard practice." Currently all experiments are based on a MATLAB implementation and the NUTS version is precisely from the published 2014 article and follows Hoffman's implemenetation. We are going to provide code that reproduces all our results.

**Reviewer 3:**
"The specific algorithm (and simplifications) for MALA are clearly the "highlight", performance-wise, but may be slightly lower impact simply due to implementation headaches (it can be done easily enough by people familiar with deep learning or AD software, but that leaves out many practicing statisticians)." Thanks for the comment. For the final version we plan to release a non automatic differentiation (AD) based MATLAB implementation for all proposed algorithms. We also plan to provide pseudo-code showing how this fast MALA scheme can be implemented with the minimum number of vector operations (few details about this are already part of the supplement) and with AD possibly used only to compute the gradient of the log target.

"the one thing I find unsatisfying is the need to fall back on existing results for "optimal" acceptance rates when tuning $\beta$. For RW and MALA proposals, maybe this is even reasonable (there's some theory about optimal preconditioners, proposal covariances, mass matrices anyway), but it would be nice to address this more generally, or at least discuss the appropriateness of falling back on these recommendations." We agree that this is a limitation. The ideal will be to automaticlaly "learn" what is the optimal average acceptance rate for a specific target. Hoewever, the standard heuristics for adapting $\beta$ worked well in the all our experiments.

"For MALA and NUTS, what sort of preconditioning is done as a baseline? For NUTS as implemented in STAN and PyMC3, I know a standard practice is to run a short pre-run which is used to estimate the mass matrix; this is then used fixed with only the step size adapted online. Is something like this done here? What about for the non-adaptive MALA?" For NUTs we use the algorithm as defined in the initial 2014 paper and it is based on the corresponding implementation of Hoffman, which does not use preconditioners. Also the non-adaptive MALA is not using a preconditioner. Notice our method regarding MALA essentially allows to obtain a full covariane preconditioner using gradient-based adaptation.

"line 140, L is described as a "positive definite lower triangular matrix" … presumably a typo (it is correct later for MALA), but LL' is positive definite, while L is lower triangular." Thank you, we will clarify this.

[Meta-Review · NeurIPS 2019]

Proposes methods to update proposal distributions for MCMC using variational-type bounds and showed it is competitive with state of the art MCMC methods. Reviewers agreed this was an important contribution.